# Slavery, the Hebrew Bible and the Development of Racial Theories in the Nineteenth Century

**Kevin Burrell** [1,2]

1  Department of Religious Studies, Burman University, Lacombe, AB 6730, Canada; kevinburrell@burmanu.ca
2  Research Fellow in the Department of Old and New Testaments, Matieland, Stellenbosch 7602, South Africa

**Abstract:** Racial ideas which developed in the modern west were forged with reference to a Christian worldview and informed by the Bible, particularly the Old Testament. Up until Darwin's scientific reframing of the origins debate, European and American race scientists were fundamentally Christian in their orientation. This paper outlines how interpretations of the Hebrew Bible within this Christian *Weltanschauung* facilitated the development and articulation of racial theories which burgeoned in western intellectual discourse up to and during the 19th century. The book of Genesis was a particular seedbed for identity politics as the origin stories of the Hebrew Bible were plundered in service of articulating a racial hierarchy which justified both the place of Europeans at the pinnacle of divine creation and the denigration, bestialization, and enslavement of Africans as the worst of human filiation. That the racial ethos of the period dictated both the questions exegetes posed and the conclusions they derived from the text demonstrates that biblical interpretation within this climate was never an innocuous pursuit, but rather reflected the values and beliefs current in the social context of the exegete.

**Keywords:** Hebrew Bible; Old Testament; race; racism; racial theories; race and ethnicity; Curse of Ham; Hamitic Hypothesis

## 1. Introduction

Racism, understood as prejudice based on the premise of fundamental biological differences between human groups, is a phenomenon that developed only in the modern West (Montagu 1997; Braude 2011). European imperial expansion and colonialization of foreign territories which began with Portugal and Spain in the latter half of fifteenth century were the most significant catalysts for the emergence of racial science (Mazzolini 2014; Horsman 1981). Ultimately, both the imperial prowess of Europe as well as human physical diversity found explanatory power in the concept of *race*. By the nineteenth century, race, both as a popular and a scientific concept, had come to dominate European and American thought (Gould [1981] 1996; Kidd 2006). Every cultural phenomenon could purportedly be determined based on race. The Scottish Anatomist Robert Knox epitomizes the significance that race had acquired in nineteenth century imagination when he confidently asserted:

> that race is in human affairs everything, is simply a fact, the most remarkable, the most comprehensive, which philosophy has ever announced. Race is everything: literature, science, art—in a word, civilization depends on it. (Knox 1862, p. v)

Knox's positivistic assessment of the racial phenomenon is emblematic of nineteenth century epistemology which at its core was characterized by "systematic attempts at using race as the primary or even sole means of explaining the workings of society and politics, the course of history, the development of culture and civilization, even the nature of morality itself" (Biddiss 1976, p. 245). But exactly how had race arrived at this threshold? How did this novel concept come to dominate popular imagination and spawned some of the greatest intellectual struggles of the period? What were its roots? And how had



race come to take on such a comprehensive and absolute character for Knox and his contemporaries?

The development of racism has often been framed as a scientific problem, tied to the emergence of modern biology (e.g., Stanton 1960; Gould [1981] 1996). Its origin is frequently linked to intellectual secularism or the manifestation of social Darwinism in the latter half of the nineteenth century (Hickman 2013). In challenging this view, Richard Popkin (1974b) reminds us that long before the rise of scientific racism, modern philosophy played a formative role in the manifestation of racial thinking. Emanuel Eze (1997a, 1997b) suggests the same by highlighting the development of racialist thought in eighteenth century philosophers. The racial views of David Hume, Emmanuel Kant, and others are now well documented (Popkin 1974b; Eze 1997b; Bernasconi 2001; Eigen and Larrimore 2006). Kant, for instance, has been called "the real founder of British racism" (Curtin 1964, p. 377), and Hume's most unambiguous espousal of racial doctrine has received more than its fair share of commentary:

> I am apt to suspect that the negroes and in general all the other species of men (for there are four or five different kinds) to be naturally inferior to the whites. There never was a civilized nation of any other complexion than white, nor even any individual eminent either in action or speculation. No ingenious manufacturers amongst them, no arts, no sciences . . . Not to mention our colonies, there are negroe slaves dispersed all over Europe, of which none ever discovered any symptoms of ingenuity. (citation in Popkin 1974b, p. 143; Gould [1981] 1996, pp. 72–73)

Hume's suspicion is revealing of the fact that eighteenth century philosophers were actively thinking and writing about race, thus contributing to the establishment of a racial hierarchy which aggrandized European achievement and denigrated other races, especially blacks. Beyond philosophy and biology, the development of racism has also been parsed as a political, social, or economic problem. Several scholars have urged, however, that endeavors to identify precursors of contemporary racial phenomena have often failed to adequately examine their principal forebear: theological racism underpinned by biblical anthropology (Kidd 2006; Livingstone 2008; Hickman 2013; cf. Harrison 1998). Colin Kidd (2006) makes the case for giving Christianity—"the dominant feature of western cultural life"—pride of place in the development and articulation of race as a social construct (p. 19). For the fact that Christianity and modernity are inextricably linked, Kidd identifies Christian *theology* as the progenitor of the modern concept of race. Hence, Michael Allen Gillespie's emphasis on "the central role that religion and theology played" in the birth of modernity (Gillespie 2008, p. xi), aptly applies to the formation of the idea of race. Indeed, because modern philosophy is a child of theology, even philosophy's contribution to the canonization of racial doctrine is still an element of Christian cultural discourse. Thus, rightly classed, race is a modern cultural construct with a religious foundation.

Accordingly, racial ideas which took shape on both sides of the Atlantic between the seventeenth and nineteenth centuries were forged with reference to a Christian worldview and informed by the Bible, particularly the Old Testament book of Genesis (Kidd 2006; Livingstone 2008). Up until Darwin's scientific reframing of the origins debate, European and American race scientists were fundamentally Christian in their orientation. This paper outlines how interpretations of the Hebrew Bible within this Christian *Weltanschauung* facilitated the development and articulation of a racial hierarchy which placed Europeans at the pinnacle of divine creation and at the same time legitimized some of the most virulent expressions of anti-black racism.

In particular, I seek to show how the primeval account of Genesis served as a seedbed for identity politics as its origin stories were plundered in service of articulating a racial ideology which vilified or bestialized blackness, and even sought to exclude Africans from the Genesis genealogies altogether. That the racial ethos of the period dictated both the questions exegetes posed and the conclusions they derived from the text demonstrates that biblical interpretation within this climate was never an innocuous pursuit but rather re-

flected the values and beliefs current in the social context of the exegetes. This hermeneutic, moreover, served a definitive purpose: to provide moral-theological, and even "scientific", justification for the unequal treatment and enslavement of blacks in north-Atlantic societies.

## 2. The Monogenism–Polygenism Debate

Many narratives of the Hebrew Bible demonstrate how constructions of difference embraced as divine injunction often facilitated the uncharitable treatment of the Othered— the Canaanites in the Promised Land as the obvious example. It seems inevitable then that certain stories of the Christian Old Testament would be harnessed as divine sanction for a host of abuses against Others, both within and without the Christian orbit. From the encounters of Europeans with indigenous peoples in the New World, to the trafficking and exploitation of enslaved Africans, to segregation in America and apartheid in South Africa, the Christian Old Testament has served as a legitimizing totem for human malignity. The dispossession of the indigenous peoples of North America by settler colonists, for example, was in many ways a reification of the ancient Israelite–Canaanite conflict (Cave 1988; Newcomb 2008; Newman 2016). Indigenous "pagans" could be driven out of the Promised Land by New Israelites wielding Old Testament tales of providential conquests. In terms of the genealogy of western racism, however, it is the origin stories of the Hebrew Bible which were central to the formation and transformation of nineteenth century racial discourse.

Like the other Abrahamic faiths (Judaism and Islam), Christian cosmology was predicated on a literalistic acceptance of the Genesis account of human origin and descent. The doctrine that biblical Adam and Eve were the foreparents of all human beings was axiomatic to a Christian conception of salvation history (Livingstone 2008). However, the astounding variety of human physiological differences which were being widely documented by natural scientists in the Age of Expansion raised questions that European intellectuals of all caliber were forced to confront. The discovery of greater and greater human diversity seemed to defy the tripartite racial taxonomy of Shemites, Hamites, and Japhites deduced from the Mosaic record (Kidd 2006; Livingstone 2010). New speculations about human origins threatened the very foundation of the Christian worldview in which Europe had been ensconced for well over a millennium. Was the biblical story of origins, a *prima facie* doctrine of Christianity, to be taken literally? Were people groups displaying such widely diverging somatic characteristics descended from the singular lineage of biblical Adam, or was their existence to be accounted for by some other means?

These questions would eventually divide western anthropology into two schools of thought. *Monogenism* and *polygenism* were the ideological camps within which intellectual battles for the right to explain human origins and classify human physical diversity were fought (Gould [1981] 1996; Livingstone 2008). These two schools were to define the debate on human origins for some three centuries before eventually being eclipsed by the Darwinian revolution in the latter part of the nineteenth century. The intellectuals espousing monogenism and polygenism were the leading scientists of the age, pioneering new fields such as natural history, geology, physical anthropology, ethnology, among others; and it is their cogitations about race that first gave definitive shape and form to what was, prior to the eighteenth century, only a penumbra (Fredrickson 2002). Monogenism, the established orthodoxy on human origins, maintained that all mankind had descended from a single set of ancestors roughly four thousand years ago. This belief laid at the very foundation of Christian theology and was virtually unquestioned prior to the middle of the seventeenth century (Kidd 2006).

Polygenism, on the other hand, proposed multiple origins as the best solution to human diversity, but its exponents would lurk in the shadows for centuries, as few had the temerity to openly advocate views which were seen as heretical and subversive to Christian orthodoxy (Livingstone 1992, 2008). Monogenists, committed to upholding the authority of scripture, condemned polygenesis as a pernicious heresy and vigorously defended the brotherhood of the human race (Kidd 2006; Livingstone 2008).

*2.1. Monogenism and the Racial Order*

Monogenists proposed environmental adaption as the primary vehicle driving human variation. This explanation harks back to antiquity. In neoclassical cosmology, the environmental argument provided a facile rationale for human diversity (Harrison 1999; Douglas 2008; Jordan 2012). Writers like Herodotus in the 5th century B.C. posited the heat of the sun as the cause of the "burnt face" of the *Aithiops*, and Roman writers alike who commented on the subject had suggested as much (Jordan 2012). Monogenists deployed the same arguments with increasing sophistication from the seventeenth to the nineteenth centuries. In their view, skin color, like other physical variation, was externally induced, superficial, and the product of climatic adaptation (Livingstone 2008).

The most obvious example of this, Africans, were black by reason of the heat from the torrid zone (Jordan 2012; Douglas 2008). Some monogenists even advanced the view that black skin could become white under ideal environmental conditions (Popkin 1974b). This idea—that black skin could become white—reveals the near-universal consensus among monogenists that white was the aboriginal color of mankind. Though there were a few notable exceptions, race theorists "took it for granted that the natural state of man is to be white and that Adam and Noah were white" (Popkin 1974b, p. 134; cf. Mazzolini 2014, p. 140). Darker skin pigmentation was explained by way of "degeneration". Non-white peoples had degenerated from the ideal color by migrating to less-than-ideal climates (Popkin 1974b). Moving beyond the symbolic association of black and white with good and evil, a relic of the medieval period (Fredrickson 2002), monogenists now associated these colors with aesthetics: white was not only the original color of Adam and Noah, but it was also the most "beautiful".

The German physician Johann F. Blumenbach (1752–1840), the leading race theorist in the eighteenth century and the "father" of physical anthropology, gave scientific credibility to both craniology (the measurement of human skulls) and the term "Caucasian" (Baum 2006; Horsman 1981). He divided humanity into five racial types: Caucasian, American, Malay, Mongolian, and Ethiopian. In Blumenbach's aesthetic evaluation the Caucasian takes pride of place for both the symmetry of the skull as well as for the beauty of white skin color (Painter 2010). "The white colour holds the first place, such as is that of most European peoples", and displays "the most beautiful form of the skull", he wrote in his treatise, *On the Natural Variety of Mankind* (Blumenbach [1775] 1795, pp. 209, 269). In the third edition of his treatise, Blumenbach famously appropriated the term *Caucasian*, based in part on "a most beautiful skull of a Georgian female", to describe both the beauty and originality of European peoples (Baum 2006, p. 77):

> I have taken the name from Mount Caucasus, both because its neighborhood, and especially its southern slope, produces *the most beautiful race of men*, I mean the Georgian; and because all physiological reasons converge to this, that in that region, if anywhere it seems we ought with the greatest probability to place the autochthones [original forms] of mankind. (Blumenbach [1775] 1795, p. 269; emphasis mine)

For Blumenbach, not only were Adam and Noah white, but their best present-day representation were Europeans, specifically the people of the Caucasus region who were reputed for their beauty (Gould [1981] 1996; Baum 2006; Painter 2010; Figal 2014). Crucially, the biblical account of Noah's ark landing on Mount Ararat (the lesser Caucasus) influenced Blumenbach's choice for the aborigines of mankind (Figal 2014). Thus, it was the temperate climate of "Europe" (or Asia?) that had served as the autochthonous homeland of the primeval, *Urstamm* (original stem) of humanity—the Caucasian. Yet the irony of Blumenbach's choice for the "fetishized female from the Caucasus" as the ideal representation of European beauty is not lost on Sara Figal (2014) who wryly comments: "Travel writers of the seventeenth and eighteenth century (unreliable, yet widely read) identify the most beautiful women in the world as the 'primitive' Georgians and Circassians from the Caucasus mountains, additionally noting their high value on the Ottoman slave market" (p. 163). Absurdly, it was the image of the Georgian female slave which became "the

unlikely icon for racial theorists", and "the genealogical source of the European race" (Figal 2014, pp. 163, 165; cf. Baum 2006; Painter 2010). Blumenbach's aesthetic racial schematization was adopted far and wide and helped to cement the color-coded racial hierarchy in European and American consciousness.

If white was the ideal color, degeneration theorists held that the farther skin color deviated from white the uglier it became (Popkin 1974b). Christoph Meiners, Blumenbach's colleague, identified two races based on color aesthetics: "white and beautiful", and "dark-skinned and ugly" (Figal 2014, p. 175). Black skin was not only deemed the ugliest, but its possessors were said to be the farthest degenerated from primeval humanity. Though still regarded as part of the human family, blacks were its worst representation, often portrayed as barely above the simian in form and intelligence. The French naturalist Georges Cuvier (1769–1832), a monogenist and contemporary of Blumenbach, spoke for many when he remarked that the Black was "the most degraded of human races, whose form approaches that of the beast and whose intelligence is nowhere great enough to arrive at regular government" (Cuvier 1997, p. 105). The bestialization of blackness was widespread among raciologists of both stripes and would only intensify as racial science gained momentum.

To be sure, the degeneration argument based on environmentalism was central to monogenetic explanations of human divergence (particularly during the eighteenth century), not only because it maintained the essential unity of mankind, but also because it left open the possibility for the moral and spiritual improvement of degenerate types by means of Christian conversion. The Negro may be the most degraded and poorest specimen, but as part of the human brotherhood he could still reap the spiritual benefits of the race (i.e., redemption; Gossett 1997). The defense of human unity by monogenists was not born out of altruistic concern for the Negro, however; it was the greater interest to uphold scriptural authority that compelled monogenists to view humanity as a brotherhood (Kidd 2006). The Lutheran clergyman and ardent monogenist John Bachman (1790–1874) could thus affirm in 1850 that the African was "of our own blood", yet still maintain that "in intellectual power" he was "an inferior variety of our species" (in Gossett 1997, p. 63).

Nevertheless, the claim of brotherhood seems to have had at least one unintended consequence. According to Fredrickson (2002), the "orthodox Christian belief in the unity of mankind, based on the Bible's account of Adam and Eve as the progenitors of all humans, was a powerful obstacle to the development of a coherent and persuasive ideological racism" (p. 52). That obstacle could not long hold, however, for polygenists were not satisfied to simply affirm the organic inferiority of the Negro; they were bent on cutting all affiliation with him. In time, polygenists would remove the African entirely from the family of Adam, with some stressing the futility of evangelistic efforts by monogenists (Gossett 1997).

### 2.2. Polygenism and the Racial Order

Polygenists were fundamentally at odds with the notion of climate as a cause for human physical variation. How could climate alone be responsible for the patent differences in human physiognomy? How does one account for variation within the same latitude, such as darker pigmentation of native Indians in America compared to Europeans, and "Eskimos" with olive skin living in the northernmost zones? And what about the evidence from centuries of Christian colonization indicating the permanence of skin color in new environments? For these and many other reasons polygenists eschewed altogether with the fundament of climate. Inherent and permanent biological differences best explained human variation, they averred. They advanced the idea of separate and distinct creations as the cause for the striking distinctions in human appearance.

The "father" of polygeny Isaac La Peyrère, a French Calvinist of Jewish descent who later converted to Catholicism, published his seminal treatise on polygenist anthropology, *Prae-Adamitae*, in 1655. Radically departing from established orthodoxy, La Peyrère argued that Adam was not the first man to be created; he was merely the father of the Jews. The Genesis account was therefore not a universal history of mankind but rather the specific

theological history of the Jews. La Peyrère introduced the idea of "pre-Adamites", or "men before Adam" as the best explanation for the variety of mankind existing on the earth (Popkin 1974a; Livingstone 1992, 2008). Peyrère's challenge to Christian orthodoxy raised the possibility that all human beings were not of "one blood" after all, but that the preadamites were in fact a different creation altogether, somewhere above the beasts but not quite as "human" as Adam, the first Jew (Livingstone 2008; Fredrickson 2002).

Moreover, for La Peyrère, preadamism helped to resolve many of the perplexing enigmas in the foundation stories of the Hebrew scriptures, such as where Cain got his wife (Peyrère found the incest option for the initial peopling of the earth less than satisfying), the populating of the city he built, and the fact that he was afraid that he would be killed by hostile hands (Livingstone 2008). Though La Peyrère's heresy was summarily condemned and vigorously refuted in the subsequent centuries, his preadamite theory would live on, being variously picked up by adherents until its flowering in the middle decades of the nineteenth century (Kidd 2006). In time, Peyrère's Adam would become the father of the *Caucasian* race (and in some iterations to the exclusion of the Jews!), and his preadamites the progenitors of the black races. Biblical history too, would in due course become Caucasian history, to the exclusion of the "non-historical" races.

Notwithstanding the survival of polygeny, the notion that there were separate creations posed too radical a challenge to Christian belief in original sin and redemption (Kidd 2006). Defenders of Christian orthodoxy rigorously attempted to stamp out the isolated embers of polygenetic heterodoxy lest they should become fires. In the middle decades of nineteenth century, however, and particularly among intellectuals of the American South committed to the defense of black enslavement, the sparse flames of polygenism would become a conflagration (Livingstone 2008; Kidd 2006). Advocates of multiple origins, particularly the naturalists of the American School of Anthropology, would marshal ever more sophisticated arguments from craniology, phrenology, zoology, geology, archaeology, and even from scripture to buttress their views of separate and distinct creations. Although Darwinian evolution in time would permanently change the course of the origins debate, nineteenth century Christians would appeal increasingly to polygenetic arguments in support of virulent racial bigotry (Livingstone 2008).

### 3. The Formidable Influence of American Polygeny

Though Polygenism first appeared in seventeenth century Europe, it was in the United States among the anthropologists, biologists, archaeologists, geologists, and other natural scientists of the formidable American School of Ethnology that it would achieve its greatest influence (Stanton 1960; Gould [1981] 1996). More than any other body of intellectuals, these scholars were responsible for transforming the "heresy" of polygenesis into orthodoxy. As Gould writes, polygenesis "was one of the first theories of largely American origin that won the attention and respect of European scientists" (Gould [1981] 1996, p. 74). Polygenists of the American School enthusiastically embraced the views of La Peyrère and "celebrated" him as a heroic martyr of science and free thought (Livingstone 2010, p. 208).

The reputed founder of the American School, Samuel Morton (1799–1851) took the scientific measurement of skulls where none had gone before him. By the time of his death, he was in possession of the largest skull collection in the world containing hundreds of skulls, appropriately called the "American Golgotha" (Gossett 1997, p. 58; Livingstone 2008, p. 174). His aptitude for meticulous measurements and collection of extensive data describing cranial capacity quickly distinguished him as a reputable authority on race science. His racial scheme which firmly secured the place of Europeans at the top and that of Africans at the bottom, was legitimized by "statistical measurements, visual imagery, and . . . moral cartography" (Livingstone 2008, p. 175).

In his *Crania Americana* (1839) and his *Crania Ægyptiaca* (1844), Morton advocated the permanent and immutable inferiority of the American Indian and the Negro, respectively. His certitude that the Negro was "an entirely different species" and the "lowest grade" of the races was buttressed by voluminous statistical measurements (Gossett 1997, p. 59).

Though Morton's statistical "finagling" has been scrutinized and refuted by Stephen J. Gould ([1981] 1996), Morton nonetheless had established the skulls of Europeans to be, by every measure, the largest in size, cranial capacity, and facial angle, and those of Negroes to be the smallest by the same measures (Livingstone 2008; Dain 2002). Mortonian polygeny had an enormous influence at home and "was spreading like wildfire" abroad (Livingstone 2008, p. 142). By the time of his death in 1851, he had "convinced most of the scientific community" of the statistical validity of his craniometric measurements for determining racial essences (Gossett 1997, p. 63). Not surprisingly, the Southern slaveholding plantocracy lapped up his findings with alacrity. Morton was so highly regarded by this coterie of slaveholders that, in 1851, the *Charleston Medical Journal* eulogized him thus: "We of the South should consider him as our benefactor, for aiding most materially in giving to the negro his true position as an inferior race" (Gibbes 1851, p. 597; in Stanton 1960, p. 144).

Though Morton advanced a "secular preadamism", which attempted to divorce science from religion (Livingstone 2008, p. 174), he nevertheless maintained that his "theory of polygenic origin was not inconsistent with the Bible" (Gossett 1997, p. 63). Multiple origins, he believed, was in harmony with "the sublime teachings of Genesis" (Stanton 1960, p. 142). Moreover, Morton echoed La Peyrèrian preadamism by supporting the idea that the Garden of Eden was "a paradise for the *Adamic race*" and not a "collective centre for the whole human family" (citation in Stanton 1960, p. 142). For Morton, the Genesis creation story represented only one branch of the human race—the Adamic branch—while omitting specific mention of the others (Stanton 1960).

The eminent Swiss-born Geologist Louis Agassiz, who together with Morton is said to be "the two most famous advocates of polygeny" (Gould [1981] 1996, p. 74), had emigrated to the United States in the 1840's and had become a strong supporter and colleague of the American School. He too defended their polygeny on scriptural grounds. In his article, "The Diversity of Origin of the Human Race", appropriately published in *The Christian Examiner*, he railed against "the charge so often brought against us", namely, "that we have undertaken to undermine our sacred books, to diminish their value, and to derogate from their holy character". To the contrary, Agassiz averred, "we deny that, in the views which we take of these questions, there is anything contradicting the records in Genesis" (Agassiz 1850, p. 111). Though Agassiz claimed that mankind shared a "spiritual and moral unity" he nonetheless maintained, like Morton, that blacks had a separate origin from whites, and that Genesis is a history of only the Caucasian race (Kidd 2006, p. 141). According to Agassiz, "the history in Genesis", outlines only "the branches of the white race"; the "colored races" or "non-historical" races are "nowhere" alluded to (Agassiz 1850, p. 111). Rather, the colored races originated in the various places where they are found.

Decidedly more anti-clerical than Agassiz their friend and colleague, Morton's two most ardent disciples, the physician Josiah C. Nott (1804–1873) and the Egyptologist George R. Gliddon (1809–1857) vigorously championed the polygenic origin of mankind, the denigration of the black race, and the view that Genesis (albeit allegorical) was solely about the history of the white race. In 1854, they published their momentous *Types of Mankind,* a tome of eight hundred pages which achieved instant success. By 1860, their bestseller was into its eighth edition and became "the leading American text on human racial difference" (Gould [1981] 1996, p. 68; cf. Gossett 1997; Young 1994). Nott and Gliddon's agenda was unapologetically pro-slavery, anti-clerical, and political (Gossett 1997; Dain 2002). They attacked their clerical opponents as unscientific, superstitious, and theologically prejudiced, and set out to writing a purely scientist account of racial typology untrammeled by theological constraints (Gossett 1997; Livingstone 2008).

Throughout their voluminous work they appealed extensively to the new science of Egyptology. Drawing on ancient Egyptian art and iconography, they sought to prove that not only biblical chronology, but also biblical anthropology was inaccurate. The monuments of ancient Egypt, they asserted, proved beyond any reasonable doubt that differentiated types of mankind were as old as creation itself. Each racial type also demonstrated a "consequent permanence of moral and intellectual peculiarities", which affirmed their

proper place on the "social scale" of "Providence" (Nott and Gliddon 1854, p. 50). In their hierarchy of creation, "God's noblest work" is the Caucasian, while the "'slough of despond' in human gradations" can be found in Africa (Nott and Gliddon 1854, p. 191).

The basic polygenic arguments of Nott and Gliddon (and the American School generally) was that nature had permanently fixed the Negro's place, as it did for all other races, and hence any attempt to alter what was fixed by supposing the inferior types could change their station was misguided. As a result, the pernicious notion being increasingly agitated by abolitionists at home and abroad that the Negro race could be elevated through emancipation and education was not only altogether futile and a waste of resources but was in fact an attempt to "arraign Providence" (Nott and Gliddon 1854, p. lii). In defense of the permanent racial stasis of blacks they claimed:

> The monuments of Egypt prove, that Negro races have not, during 4000 years at least, been able to make a solitary step, in Negro-Land; the modern experiences of the United States and the West Indies confirm the teaching of the monuments of history; and our remarks . . . hereafter, seem to render fugacious all probability of a brighter future for these organically-inferior types. (Nott and Gliddon 1854, pp. 95–96)

Consequently,

> [I]t would seem that the Negroes . . . must remain substantially in the same benighted state wherein Nature has placed them and in which they have stood, according to Egyptian monuments, for at least 5000 years. (Nott and Gliddon 1854, p. 189)

According to these architects of human worth nothing but "a miracle"—the "silliest of desperate suppositions"—could possibly change the Negro's inherent inferiority (Nott and Gliddon 1854, p. 191). Again, the thesis of the organic and permanent fixity of black inferiority served the intents and purposes of slave owners with exceptional force. Indeed, Nott and Gliddon gave explicit sanction to the southern industry by writing that "the physical characteristics of a 'field,' or agricultural, 'Nigger' were understood at Rome 1800 years ago, as thoroughly as by cotton-planters in the State of Alabama, still flourishing in A.D. 1853" (Nott and Gliddon 1854, p. 252). With such unequivocal endorsements, little wonder southern states embraced these "scientific" arguments with enthusiasm and provided generous financial support for such research. Indeed, because slave money funded many academic institutions and disciplines, for obvious reasons the academy was obliged to overtly endorse or give an implicit nod to the industry through its silence. Along the same lines, several scholars have shown the connection between slavery and the growth of modern industry—from the Industrial Revolution to capitalist enterprises in Europe and America—suggesting that it was this web of economic entanglement which permitted the perpetuation of slavery for as long as it did (Williams 1966; Baptist 2014). Scripturalists as well, many of whom were slave owners themselves, were no less willing to do violence to the scriptures for economic reasons.

Another weapon in Nott and Gliddon's ethnological crusade involved the use of visual representation. To further bolster their claim of black inferiority, they produced a significant number of visual aids such as maps and sketches to show the close associations between the Negro and the ape (see Figure 1 for one such example). Still, they were not content to rely solely on scientific arguments and visual cartography to establish once and finally the permanence of black inferiority; as we shall see momentarily, they would carry forward a new and bold exegesis of the biblical text itself.

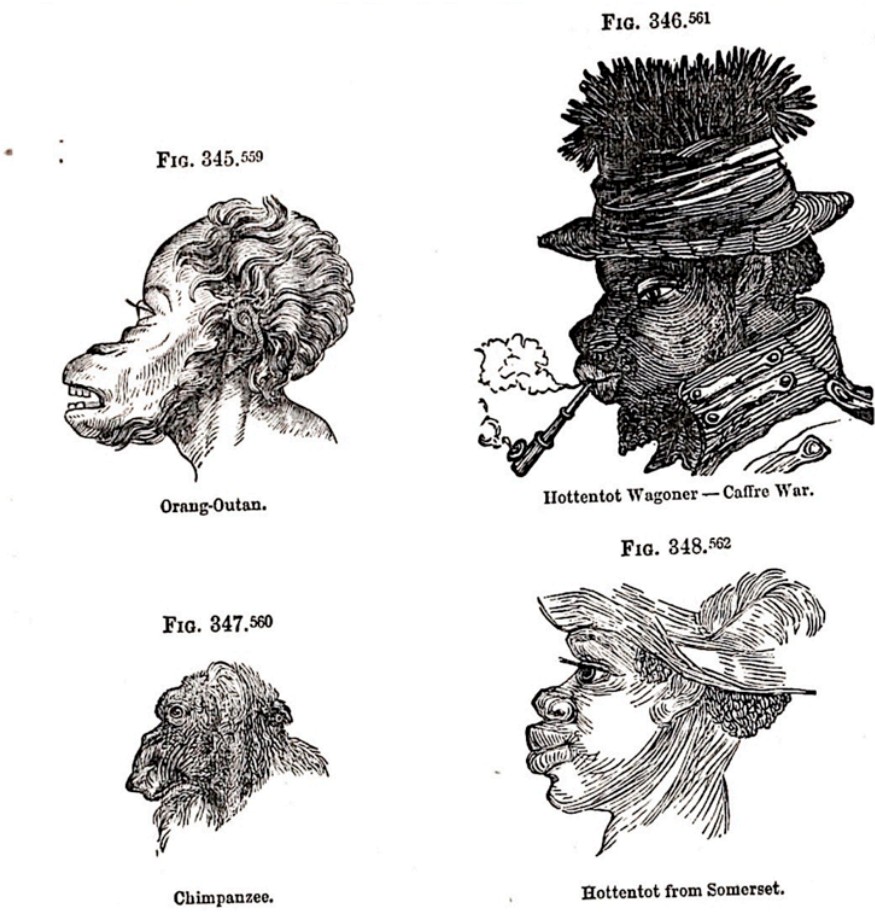

**Figure 1.** Comparison between blacks and apes; *Types of Mankind*, p. 459.

### 4. The Bible and Slavery

Monogenists, as we have seen, stressed the unity of the human race. However, slavery as an institution thrived concurrently with unctuous professions of human unity. How, then, did monogenists justify the enslavement of Africans, their brothers? Inasmuch as scripture was always at the forefront of debates about human origin and unity, so too scripture lay at the heart of disputes regarding slavery (Haynes 2002; Whitford 2009). As already noted, it was customary for monogenists to affirm the unity of the human race based on Christian orthodoxy on the one hand, and yet still defend the inferiority of other races, especially blacks, on the other.

John Henry Hopkins (1792–1868), a northern Episcopal bishop and staunch defender of slavery wrote, "The Scriptures show me that the negro like all other races, descends from Noah, and I hold him to be a man and a brother. But though he be my *brother* it does not follow that he is my *equal*". To this end he inquires rhetorically, "why should not the African race be subject, and subject in that way to which it is best adapted?" Slavery, not freedom, Hopkins contends, is the proper place for his Negro brother (Hopkins 1864, p. 32; *emphasis original*). Similarly, Bachman, the Lutheran minister and slaveholder, whose views on black inferiority we have already encountered, was a blistering critic of Samuel Morton's polygeny. He argued vociferously in favor of monogenism and the brotherhood of humanity, and yet at the same time believed that the Negro was justly enslaved for his own benefit, like a child in need of "protection and support" (Gossett 1997, p. 63). Clearly, Bachman and other scripturalists who benefited directly or indirectly from the enslavement of Africans found it of no consequence to deploy scripture in defense of black inferiority.

It is to be noted, however, that not everyone in nineteenth century America who agreed with the notion of innate black inferiority was a supporter of slavery. Indeed, some proponents of otherwise overtly racist views of Africans were staunchly opposed to slavery.

The English surgeon Charles White, a polygenist and strong proponent of black inferiority, for example, believed that even "men of inferior capacities" (i.e., blacks) were as entitled to freedom as any Teuton (Stanton 1960, p. 18; cf. Curtin 1964; Hudson 1996). Some could declaim the inferiority of blacks with aplomb, and in the same breath vehemently denounce slavery. The pioneering American archaeologist, Ephraim George Squier epitomized this dichotomy in stating that he had a "precious poor opinion of niggers, or any of the darker races", yet he had "a still poorer one of slavery" (in Stanton 1960, pp. 192–93). That said, it would be hard pressed to find an advocate of slavery who did not also seek to justify black inferiority.

The Bible itself became a tool in the defense of slavery, especially from the sixteenth century when "for the first time in the history of mankind, the Europeans introduced a system of color-based slavery" (Mazzolini 2014, p. 146). Defenders of the institution wielded biblical accounts of slave regulations found in the Pentateuch, and even drew upon New Testament examples for the same purpose (Kidd 2006). By far, however, the most important biblical justification for race slavery was the protean myth of Ham's Curse (Braude 1997, 2005; Goldenberg 2003, 2017).

### 4.1. The Hamitic Hypothesis

Clerical monogenists had long favored the Hamitic myth, rooted in the exegetical distortion of Genesis 9: 18–27, to justify black inferiority and servitude. While forms of the myth had circulated in early Christian, Jewish, and Muslim literature, no coherent or systemic association of Ham with blackness and servitude had developed prior to the late eighteenth century (Goldenberg 1997; Whitford 2009; Aaron 1995). Kidd (2006) shows that Ham was not primarily associated with skin color in the medieval period, but rather stood as a symbol of idolatry and polytheism (p. 75). From another perspective, Benjamin Braude (2005) demonstrates that the Hamitic Hypothesis has been amazingly fungible, shifting its form to incorporate various "descendants" of Ham, depending on the social context and object of vilification. According to Braude, it was the image of the Jew that was most frequently associated with the benighted son of Ham in the medieval period. However, this would change dramatically in the nineteenth century: the myth of Ham would acquire "its most notorious manifestation in antebellum America" (Goldenberg 2017, p. 1), whereas "The thirteenth century depiction of Ham makes him a Jew. The nineteenth century depiction of Ham makes him a Black" (Braude 2005, p. 80). This shift was certainly born out of a need to justify slavery in antebellum America, but it was also specifically the result of concerted exegetical maneuverings of the biblical story.

In 1754, the English Bishop Thomas Newton (1704–1782) penned "one of the most important defenses of the Curse of Ham ever written" (Whitford 2009, p. 141). The decisive and influential publication, *Dissertation on the Prophecies*, would have both immediate and far-reaching consequences for the exegesis of Genesis 9: 18–25. David Whitford (2009) has indicated that this seminal *Dissertation*, above all else, influenced later articulation and promulgation of the myth of Ham's curse in a host of publications, "from exegetical works, to abolitionist parodies of proslavery tracts, to the *Encyclopedia Britannica*, to the United States Congressional Record" (p. 160). Exegetically manipulating the text, Newton "corrected" the reading of Genesis 9: 25 to unambiguously place the curse on Ham instead of on Canaan, and he affirmed the prophetic fulfillment of the curse in the enslavement of Africans. According to Newton, the curse upon Ham and his posterity is an "extraordinary prophecy", "both wonderful and instructive" which has been "shown to be fulfilled from the earliest times to the present" (Newton [1754] 1832, pp. iii, 14). He explained the historical fulfillment thus:

> The whole continent of Africa was peopled principally by the children of Ham; and for how many ages have the better parts of that country lain under the dominions of the Romans, and then of the Saracens, and now of the Turks! In what wickedness, ignorance, barbarity, slavery, misery, live most of the inhabitants! And of the poor negroes how many hundreds every year are sold and bought

like beasts in the market, and are conveyed from one quarter of the world to do the work of beasts in another! (Newton [1754] 1832, p.12)

Newton hesitantly attributes the blackness of Hamites to the curse. He writes, "We might almost as well say (as some have said) that the complexion of the blacks was in consequence of Noah's curse" (Newton [1754] 1832, p. 12). Subsequent to Newton, theologians of all stripes affirmed his exegetical conclusions, and were not in the least hesitant in associating black skin color with the curse of Noah. By the nineteenth century, the literature propounding the blackness of Ham was voluminous. For these exegetes the prophecies of Noah had been fulfilled with exactitude.

Hopkins, the Episcopal bishop of Vermont whom we have already met, was a prominent defender of slavery. In 1864, he wrote an exhaustive outline of biblical evidence in support of slavery entitled *A scriptural, ecclesiastical, and historical view of slavery.* Of import, Hopkins appeals to "Bishop Newton, whose well-known work upon the Prophecies" is "selected by the Church for students in Theology" and therefore "a safe guide of ministerial opinion" (Hopkins 1864, p. 70). Hopkins sanctioned the justice of Southern slavery by appealing to the curse of Noah: "the Deity pronounced the curse of slavery upon the posterity of Ham" resulting in "the total degradation of Ham, in the slave-region of Africa" (Hopkins 1864, pp. 67, 69). The curse against Ham, he suggested, had been fulfilled with tremendous accuracy: "And all history proves how accurately the prediction has been accomplished, even to the present day" (Hopkins 1864, pp. 7, 19). On the moral question of slavery, he concludes, "there was necessarily no sin whatever" (Hopkins 1864, p. 45).

Similarly, the clergyman Josiah Priest published his influential *Slavery, as It Relates to the Negro* in 1843 filled with theological arguments in favor of black slavery. Appropriately giving deference to "Bishop Newton" on the exegesis of Ham's tale, Priest adjudicated the morality of slavery thus: "wherefore we come to the conclusion, that it is not sinful to enslave the negro race" (Priest 1843, pp. 86–87; cf. Whitford 2009, p. 164). Though nearly universally defended by Southern clergy, the Curse of Ham would undergo yet another radical metamorphosis in the latter half of the nineteenth century, thanks to the restless efforts of the two firebrands of the American School.

*4.2. The New Hamitic Myth*

As we have seen, Nott and Gliddon were bent on refuting the unity of the human race by resorting to scientific arguments. However, the myth of Ham's blackness stood as a formidable obstacle to their agenda. They decided to attack the problem at the source: they would demonstrate that the biblical text bore no such evidence of Ham's curse. In a word, they would seek to loosen Newton's exegetical stranglehold on the story of Ham. Another significant point impinging upon Ham's blackness had to do with the fact that the new field of Egyptology was revealing ancient Egypt to be a highly sophisticated civilization; one which many believed at the time was the forerunner of western civilization (Young 1994). However, because the curse of Ham was predicated on the idea that *all* Hamites, including the ancient Egyptians, were black—and therefore incapable of high civilization—there was a particular urgency to set the record straight. "The immediate solution", as Robert Young (1994) points out, "was to make Egypt white" (p. 159).

The American School launched a sustained challenge to the long-established ecclesiastical position popular among monogenists that Ham was cursed with black skin, and they set about to reclaim Ham for the white race. The American School expended no little effort—through lectures and publications—to reform the blackness of the ancient Egyptians and hence to "to prove beyond possible doubt, that the Ancient Egyptian race were Caucasians" (Nott 1844, p. 16; cf. Young 1994; Trafton 2004; Bernal 1987). As for the notion that Hamites had been cursed with black skin, they engaged the biblical account directly. Providing a necessary corrective to the Genesis 9 story, Gliddon claimed that Canaan,

was not *physically* changed in consequence of the *curse*. He ever remained a *white* man, as did, and do, all his many descendants. No scriptural production can



be found, that would support an hypothesis so absurd, as that, in consequence of the curse, Canaan was transmuted into a negro ... If then with the curse branded on Canaan, and on his whole posterity, the Almighty did not see fit to change his skin, his hair, bones, or any portion of his physical structure, how unjust, how baseless is that theory (unsupported by a line of Scripture, and in diametrical opposition to monumental and historical testimony), which would make Canaan's immediate progenitor, Ham, the father of the Negroes! or his apparently blameless brother, Mizraim, an Ethiopian! (Gliddon 1850, p. 41; emphases original)

Taking their exegesis to its logical conclusion, Nott and Gliddon affirmed, as did Morton and Agassiz, that the author of Genesis "omits *Negro* races altogether, from his tripartite classification of humanity under the symbolic appellatives of 'Shem, Ham, and Japheth'" (Nott and Gliddon 1854, p. 249). Not only are Negroes never mentioned in the Pentateuch, but they would further claim that "the *Negro* races are never alluded to in ancient Jewish literature" and that "Ethiopia" was "a false interpretation of the Hebrew KUS*h*, which always means Southern Arabia, and nothing but the Cushite-Arabian race" (Nott and Gliddon 1854, p. 253). Morton too had claimed that the ancient Ethiopians had "no affinity, even in the remotest times, to the Negro race" and that "The valley of the Nile, both in Egypt and in Nubia, was originally peopled by a branch of the Caucasian race" (Morton 1844, pp. 43, 65). His disciples affirmed both with even greater zeal. Consequently, not only were the ancient Egyptians reclaimed for the white race, but even the ancient Ethiopians (Cushites) were brought into the Caucasian fold, effectively removing Negroes from ancient Jewish literature, and completely dislodging them from the environs of ancient Near Eastern civilizations. As I have written elsewhere, "under the polygenists of the American School, the long-standing association of blackness-African-slavery-Hamites was spun upon its head" (Burrell 2020, p. 49).

This "new" exegesis of white Hamites and Cushites would be picked up and reflected in a host of disciplines like Egyptology, archaeology, anthropology, ancient Near Eastern history, and of course, biblical scholarship (cf. Bernal 1987; O'Connor and Reid 2003). Thus, when in 1891 the British Orientalist Archibald Sayce wrote his influential *Races of the Old Testament* he asserted that biblical Cush or Ethiopia was "inhabited for the most part by a white race whose physical characteristics connect them with the Egyptians" (Sayce 1891, p. 51). Likewise, based on biometric analysis of ancient Near Eastern skulls, the celebrated Egyptologist and Archaeologist Flinders Petrie reached the "simple conclusion that North Africa, Egypt, and Syria were occupied by allied tribes of a *European* character" (Petrie 1901, p. 250). By the time the great British Ethnologist Charles G. Seligman wrote his *Races of Africa* in 1930 he could claim with gravity that "the civilizations of Africa, are the civilizations of the Hamites" (Seligman 1930, p. 96).

Furthermore, once American polygenism took root in the second half of the nineteenth century, a host of theological works on preadamism began to appear, including the 1860 publication of Isabelle Dunkan's *Pre-Adamite man: or the story of our old planet and its inhabitants as told by scripture and science*; Paschal B. Randolph's *Pre-Adamite Man: Demonstrating the Existence of the Human Race* in 1863; Dominick McCausland's *Adam and the Adamite; or, the harmony of scripture and ethnology* in 1864; and Alexander Winchell's *Preadamites; or a demonstration of the existence of men before Adam* in 1878. No longer in need of Ham's curse to justify black slavery, revisionist preadamism found new exegetical grounds to achieve the same ends. Preadamite advocates assaulted the Genesis origin stories with abandon in order to bestialize blacks and demonstrate black inferiority, and especially to disparage miscegenation between blacks and whites in view of the new reality of an emancipated black population of some four million (Livingstone 2008).

Black preadamites could appear as the cursed Cain and his progeny, or the fallen, miscegenating *Nephilim* of Genesis 6; and even stranger yet, the *nachash*—the crafty serpent that engaged in miscegenetic sexual temptation with Eve (Kidd 2006; Livingstone 2008; Stokes 1998). The cursed Ham was now replaced by the cursed *nachash*, or the cursed Cain,

or the Cursed *Nephilim*, or even cursed Nimrod—the exegetical outcomes seemed endless. As Livingstone (2008) writes, "The power of pre-adamism, in one form or another, to serve the interests of race hatred and the language of gross racial abuse evidently acquired a considerable following during the second half of the nineteenth century . . . the basic thrust of the scheme was to bestialize the African and to provide warrant for a fixation with blood purity" (p. 197). Here again, the Genesis stories provided fodder for racial othering, exclusion, and oppression.

## 5. Conclusions

The foregoing has demonstrated that the origin of racism was first and foremost a scripture problem. The earliest efforts to define a racial hierarchy were carried forward by European and American intellectuals espousing one of two theistic explanations of human origins. Monogenists and polygenists sought to define human origin and filiation with reference to sacred anthropology—whether affirming or contesting it. Though they vehemently disagreed in their views regarding Mosaic cosmology, both groups employed scriptural interpretation towards invidious ends. As we have seen, the establishment of a racial order which placed Europeans at the very top and blacks at the bottom rested on distorted and self-serving interpretations of the origin stories of Hebrew Bible. The vilification and bestialization of blackness, whether on biblical or "scientific" grounds, served the ultimate purpose of justifying black enslavement and oppression. Thus, from its theological origins to our contemporary arena, the tale of race has revealed the vicissitudes of human hatred and malice. That story is still playing out and its denouement remains uncertain.

To be sure, the monogenism–polygenism dialectic has left an indelible stamp on "man's most dangerous myth"; and so too did interpretations of the foundation stories of the Hebrew scriptures. Recognition of this modern history of interpretation should give pause to anyone who regard care for his fellow human as a sacred duty. That the foundation book of Judaism and Christianity could be so grossly misappropriated in the service of racial hatred and identity politics, shows that far from being an innocuous exercise, interpretations of the Bible are often commentary on the values and commitments of the wider society. This was true in nineteenth century racial politics, and it remains true in twenty-first century racial politics—if the recent and ongoing racial reckoning is any indication. The faithful exegete will responsibly appropriate sacred traditions.

**Funding:** This research received no external funding.

**Institutional Review Board Statement:** Not applicable.

**Informed Consent Statement:** Not applicable.

**Data Availability Statement:** Not applicable.

**Conflicts of Interest:** The author declares no conflict of interest.

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
