# Peer review of "Slavery, the Hebrew Bible and the Development of Racial Theories in the Nineteenth Century"

_religions, doi:10.3390/rel12090742_

Round 1

Reviewer 1 Report

It is a very well-written essay on the interconnection between biblical interpretation and racial discourse in modern history. Any reader interested in identity politics in the realm of religion would learn a lot from this work.

Author Response

Dear Reviewer,

Thank you kindly for taking the time to read and providing helpful feedback on my work. 

With gratitude!

Reviewer 2 Report

This paper is interesting and fairly well-written. I enjoyed reading it. However, I learned very little because the paper contributes no new insights to the study of interrelationships among slavery, Biblical interpretation, and racial theories in the 19th century. For the most part, the paper reviews well-worn issues in the development, in the American South and some quarters of England, of ideological justifications for the trans-Atlantic slave trade and plantation slavery. The paper, therefore, covers material that will be very familiar to scholars in history and anthropology (see, e.g., Robert Sussman's book, The Myth of Race [Harvard Univ Press 2014]).

The author misses the point that slavery and colonialism – or more accurately, the economic gains from slavery and colonialism – were the driving forces of “scholarly” efforts to bend the Judeo-Christian scriptures into a treatise that supported or excused the enslavement of Africans and the exploitation and oppression of peoples around the world. For instance, the author says virtually nothing about the connections between (1) slave traders and slave owners and (2) the early 19th century “race scientists.” Further, the author all but ignores the fact that abolitionists in the U.S. and England, most notably, William Wilberforce, developed their rationale for ending slavery by citing scriptural passages that helped to rally public support for the cause of abolishing slavery world-wide (not just in the British Empire and the U.S.). Ultimately, the author's essay leads us to conclude that the Biblical account of creation had to be challenged, altered, or rejected in order for defend the idea of a natural racial hierarchy that placed Europeans (specifically, northwestern Europeans) at the top and Africans at the bottom.  But this conclusion is well-known.

The author also makes, from time to time, some highly questionable assertions. For starters, the statement that racism emerged only in the West (lines 21-23) is false. Racism also emerged in the Islamic societies that had extensive contacts with peoples in Europe, Asia, and Africa. The author's statement that “modern philosophy is a child of theology” (line 78) is also unsupported by the historical record. Indeed, modern philosophy is largely an intellectual antidote to, and a replacement for, religion/theology.

In addition, the author's review of theological pronouncements about race focuses almost exclusively on Protestant theologians based in the U.S. and England and makes no attempt to make distinctions between Roman Catholic and Protestant theologians or make distinctions among Protestant theologians in a variety of countries. Thus, the paper lacks a comparative perspective.

I caught a small typo: On line 286, “century” should be inserted between “nineteenth” and “Christians.”

In conclusion, this paper contributes no new knowledge, and its review of literature would be more appropriate for a textbook chapter than for an article in a peer-reviewed scholarly journal.

Author Response

  1. Dear Reviewer,

    Thank you for taking the time to read and providing feedback on my work. Your points of critiques are noted. I would like to present the following considerations in response to your feedback:

    1. Though you have rightly noted that the paper covers material that is "very familiar to scholars in history and anthropology,"  it is to be noted that the paper is not specifically geared towards this specialized audience. Indeed, the work is a contribution to a special issue on "The Hebrew Bible, Race, and Racism," and as such draws on insights from the history of anthropology to show how interpretations of the Hebrew Bible has contributed to racial discourse. The work is intentionally inter-disciplinary and would therefore be beneficial to scholars of the Hebrew Bible as well as a more general scholarly audience. It is largely because scholars of the Hebrew Bible have often re-inscribed to varying degrees aspects of this racialized interpretation that this special issue was warranted.
    2. Your point that there is an economic connection between slavery/colonialism and racial theories (and the desire to bend the Scriptures to support the racial hierarchy) is sustained, and I have now included a statement to this effect. Similarly, the economic connection which often exists between slave traders/owners and racial theorists is also sustained and a short note made of this. Nevertheless, my goal is not to write a comprehensive account of all the features of and/or influences upon the development of western racism, but again to narrowly focus on the way interpretations of the Hebrew Bible served to further the racial discourse.
    3. That biblical interpretation was also a key factor in anti-slavery movements in the UK and America, or that Christian groups and theologians were fundamental to the abolition of slavery is not to be denied. It is a topic that is worth studying in its own right. But I was interested in looking at how biblical interpretation facilitated the development of racial theories up to and including the 19th century, and not how the text influenced the abolition movement.
    4.  To your point that my assertion that “racism emerged only in the West (lines 21-23) is false,” kindly note that I emphasize a specific form of prejudice based on fundamental biological differences: “Racism, understood as prejudice based on the premise of fundamental biological differences between human groups…” Both authors that I cite make the case for this assertion. The historian Benjamin Braude (How Racism Arose in Europe and Why It Did Not in the Near East. In Racism in the Modern World: Historical Perspectives on Cultural Transfer and Adaptation. Edited by Manfred Berg and Simon Wendt. New York: Berghahn Books, 2011, pp. 41-64), for example, make this point both here and in other publications. Similarly, the Anthropologist Ashly Montagu makes the same argument with respect to biological racism as a systemized articulation of group prejudice. This is not to deny, as you have rightly noted, ethnocentrism, religious prejudice, cultural prejudice, and other forms of prejudices that existed in Islamic and even societies of antiquity. Some have termed these types of prejudice as “proto-racism” to distinguish them from modern racism.
    5. Likewise, that I did not distinguish between Protestant and Catholic theologians on the issue of race and slavery or distinguish between “Protestant theologians in a variety of countries,” is as a result of my narrow focus. I did not set out to write a comprehensive account of the biblical contribution to the development of racism but wished to paint the historical landscape in broad strokes.
    6. Finally, the assertion that modern philosophy is a child of theology is supported by the argument that modernity itself has theological origins (see Gillespie, Michael Allen. 2008. The Theological Origins of Modernity. Cambridge: Cambridge University Press), but also by the fact that modern philosophy was birthed in the 17th and 18th centuries within a Christian theological context (see Richard Francks, Modern Philosophy: The Seventeenth and Eighteenth Centuries, McGill-Queen's University Press, 2003). This does not deny that modern philosophy quickly emerged as an "antidote ... to religion/theology," but its origin was both the result of and response to theological antecedents (such as the Reformation and the intellectual liberation that it spawned).

Again, thank you for your insightful response. Much appreciated.

Reviewer 3 Report

This is a very good examination - well researched, clear and concise. I appreciate the balanced/historical approach to the topic and recommend that this be published as is with just an overview of any grammatical/spelling corrections from an editor. 

Author Response

(The authors gave the same response as above.)

Round 2

Reviewer 2 Report

The author has worked diligently to revise the paper, and I am mostly satisfied by the author's responses to my comments.

I think the revised discussion on lines 397-406 is basically sound and is a fine response to my comment about the economics of slavery and 19th century race science.

I continue to disagree with the author's assertion that biological racism was a modern western phenomenon -- see Dinesh D'Souza's 1996 article, "Is Racism a Western Idea?" in Christian Ethics Today (March issue).

Finally, a small typo -- line 627, "regard" should be "regards" ("...anyone who regards care...")

Overall, this is an informative article that is highly appropriate for the special issue to which the author refers.